# Closing the Generalization Gap in One-Shot Object Detection

## Abstract

Despite substantial progress in object detection and few-shot learning, detecting objects based on a single example – one-shot object detection – remains a challenge. A central problem is the generalization gap: Object categories used during training are detected much more reliably than novel ones. We here show that this generalization gap can be nearly closed by increasing the number of object categories used during training. Doing so allows us to beat the state-of-the-art on COCO by 5.4 %AP$^{50}$ (from 22.0 to 27.5) and improve generalization from seen to unseen classes from 45% to 89%. We verify that the effect is caused by the number of categories and not the amount of data and that it holds for different models, backbones and datasets. This result suggests that the key to strong few-shot detection models may not lie in sophisticated metric learning approaches, but instead simply in scaling the number of categories. We hope that our findings will help to better understand the challenges of few-shot learning and encourage future data annotation efforts to focus on wider datasets with a broader set of categories rather than gathering more samples per category.

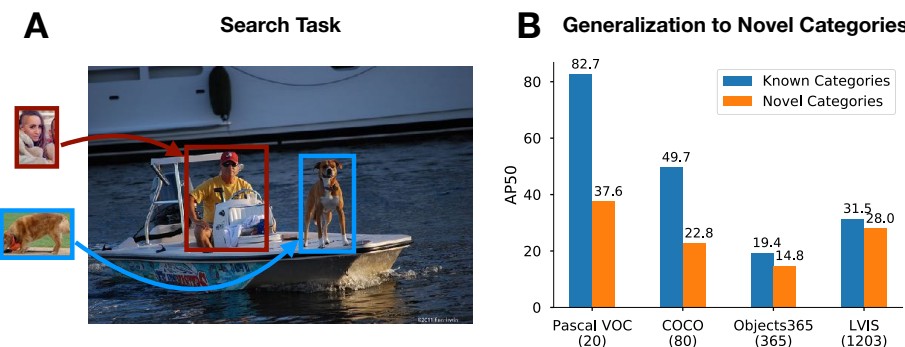

Figure 1: **A.** One-shot object detection: Identify and localize all objects of a certain category within a scene based on a (single) instructive example. **B.** Increasing the number of categories used during training reduces the generalization gap to novel categories presented at test time (in parenthesis: number of categories in each dataset).

## 1 Introduction

*It's January 2021 and your long awaited household robot finally arrives. Equipped with the latest "Deep Learning Technology", it can recognize over 21,000 objects. Your initial excitement quickly vanishes as you realize that your casserole is not one of them. When you contact customer service they ask you to send some pictures of the casserole so they can fix this. They tell you that the fix will be some time, though, as they need to collect about a thousand images of casseroles to retrain the neural network. While you are making the call your robot knocks over the olive oil because the steam coming from the pot of boiling water confused it. You start filling out the return form ...*

While not 100% realistic, the above story highlights an important obstacle towards truly autonomous agents such as household robots: such systems should be able to detect novel, previously unseen objects and learn to recognize them based on (ideally) a single example. Solving this one-shot object detection problem can be decomposed into three subproblems: (1) designing a class-agnostic object proposal mechanism that detects both known and previously unseen objects; (2) learning a suitably general visual representation (metric) that supports recognition of the detected objects; (3) continuously updating the classifier to accommodate new object classes or training examples of existing classes. In this paper, we focus on the detection and representation learning part of the pipeline, and we ask: what does it take to learn a visual representation that allows detection and recognition of previously unseen object categories based on a single example?

We operationalize this question using an example-based visual search task (Fig. 1) that has been investigated before using handwritten characters (Omniglot; Michaelis et al. (2018a)) and real-world image datasets (Pascal VOC, COCO; Michaelis et al. (2018b); Hsieh et al. (2019); Zhang et al. (2019); Fan et al. (2020); Li et al. (2020)). Our central hypothesis is that scaling up the number of object categories used for training should improve the generalization capabilities of the learned representation. This hypothesis is motivated by the following observations. On (cluttered) Omniglot (Michaelis et al., 2018a), recognition of novel characters works almost as well as for characters seen during training. In this case, sampling enough categories during training relative to the visual complexity of the objects is sufficient to learn a metric that generalizes to novel categories. In contrast, models trained on visually more complex datasets like Pascal VOC and COCO exhibit a large generalization gap: novel categories are detected much less reliably than ones seen during training. This result suggests that on the natural image datasets, the number of categories is too small given the visual complexity of the objects and the models retreat to a shortcut (Geirhos et al., 2020) – memorizing the training categories.

To test the hypothesis that wider datasets improve generalization, we increase the number of object categories during training by using datasets (LVIS, Objects365) that have a larger number of categories annotated. Our experiments support this hypothesis and suggest the following conclusions:

- The generalization gap between training and novel categories is a key problem in one-shot object detection.
- This generalization gap can be almost closed by increasing the number of categories used for training: going from 80 classses in COCO to 1200 in LVIS improves relative performance from 45% to 89%.
- The number of categories, not the amount of data, is the driving force behind this effect.
- Closing the generalization gap allows us to use established methods from the object detection community (like e.g. stronger backbones) to make further progress.
- We use these insights to improve state-of-the-art performance on COCO by **5.4** %AP$^{50}$ (from 22 %AP$^{50}$ to 27.5 %AP$^{50}$) using annotations from LVIS.

## 2 RELATED WORK

**Object detection**   Object detection has seen huge progress since the widespread adoption of DNNs (Girshick et al., 2014; Ren et al., 2015; He et al., 2017; Lin et al., 2017a; Chen et al., 2019a; Wu et al., 2019b; Carion et al., 2020). Similarly the number of datasets has grown steadily, fueled by the importance this task has for computer vision applications (Everingham et al., 2010; Russakovsky et al., 2015; Lin et al., 2014; Zhou et al., 2017; Neuhold et al., 2017; Krasin et al., 2017; Gupta et al., 2019; Shao et al., 2019). However most models and datasets focus on scenarios where abundant examples per category are available.

**Few-shot learning**   The two most common approaches to few-shot learning have been, broadly speaking, based on metric learning (Koch et al., 2015; Vinyals et al., 2016; Snell et al., 2017) and meta learning: Learn a good way to learn a new task (Finn et al., 2017; Rusu et al., 2018), or combinations thereof (Sun et al., 2019). However, recent work has shown that much simpler approaches based on transfer learning achieve competitive performance (Chen et al., 2019b; Nakamura & Harada, 2019; Dhillon et al., 2019). A particularly impressive example of this line of work is Big Transfer (Kolesnikov et al., 2019), which uses transfer learning from a huge architecture trained on a huge dataset to perform one-shot ImageNet classification.

**Few-shot & one-shot object detection**     Recently, several groups have started to tackle few-shot learning for object detection. Two training and evaluation paradigms have emerged. The first is inspired by continual learning: incorporate a set of new categories with only a few labeled images per category into an existing classifier (Kang et al., 2018; Yan et al., 2019; Wang et al., 2019; 2020). The second one phrases the problem as an example-based visual search: detect objects based on a single example image (Fig. 1 left; Michaelis et al., 2018b; Hsieh et al., 2019; Zhang et al., 2019; Fan et al., 2020; Li et al., 2020). We refer to the former (continual learning) as *few-shot object detection*, since typically 10–30 images are used for experiments on COCO. In contrast, we refer to the latter (visual search) as *one-shot object detection*, since the focus is on the setting with a single example. In the present paper we work with this latter paradigm, since it focuses on the representation learning part of the problem and avoids the additional complexity of continual learning.

**Methods for one-shot object detection**     Existing methods for one-shot object detection employ a combination of a standard object detection architecture with a siamese backbone and various forms of feature attention and concatenation on the backbone output or in the heads (Biswas & Milanfar, 2015; Michaelis et al., 2018b; Hsieh et al., 2019; Zhang et al., 2019; Fan et al., 2020; Osokin et al., 2020; Li et al., 2020). Spatially aware similarity measures (Li et al., 2020) or transformations (Biswas & Milanfar, 2015; Osokin et al., 2020) improve recognition in cases where the pose of the reference objects differs from that of the detected object. We here use one of the most straightforward models, Siamese Faster R-CNN (Michaelis et al., 2018b), to demonstrate that a change of the training data rather than the model architecture is sufficient to substantially reduce the generalization gap between known and novel categories.

**Related tasks**     A number of related pieces of work propose approaches to slightly different example-based search tasks. Examples include one-shot segmentation using handwritten characters (Michaelis et al., 2018a), natural textures (Ustyuzhaninov et al., 2018) and natural images (Shaban et al., 2017). In addition, several groups have suggested one-shot and few-shot detection tasks with slightly different focus and protocols (Dong et al., 2018; Chen et al., 2018; Schwartz et al., 2019; Wu et al., 2019a), including episodic evaluation (Wu et al., 2019a), transfer across datasets (Chen et al., 2018) and fine-grained detection (Schwartz et al., 2019). Also closely related are instance retrieval (Tolias et al., 2016) and co-segmentation (Rother et al., 2006; Hsu et al., 2019). The key difference of our work is that we do not propose a new architecture, but instead investigate the relationship between the number of categories used during training and the generalization to novel categories.

**Number of categories in few-shot learning**     Most of the few-shot learning literature focuses on developing new methods for existing benchmarks. The influence of the training data was mostly observed indirectly, e.g. through better performance on datasets with more categories such as *tiered*ImageNet vs. *mini*ImageNet. Two concurrent studies report that more categories help few-shot object detection (Fan et al., 2020) and investigate the influence of data diversity, image complexity, intra- and inter-category diversity and other factors on few-shot classification (Jiang et al., 2020). Both publications are consistent with out results that the number of categories is a key factor for improving few-shot performance.

## 3 EXPERIMENTS

**Models**     We mainly use Siamese Faster R-CNN, a one-shot detection version of Faster R-CNN (Ren et al., 2015) similar to Siamese Mask R-CNN (Michaelis et al., 2018b). Briefly, it consists of a feature extractor, a matching step and a standard region proposal network and bounding box head (Fig. 2). The feature extractor (called backbone in object detection) is a standard ResNet-50 with feature pyramid networks (He et al., 2016; Lin et al., 2017a) which is applied to the image and reference with weight sharing. In the matching step the reference representation is compared to the image representation in a sliding window approach by computing a feature-wise L1 difference. The resulting similarity encoding representation is concatenated to the image representation and passed on to the region proposal network (RPN). The RPN proposes a set of bounding boxes which potentially contain objects. These boxes are then classified as containing an object from the reference class or something else (other object or background). Box coordinates are refined by bounding box regression and overlapping boxes are removed using non-maximum suppression.

We additionally developed Siamese RetinaNet, a single-stage detector based on RetinaNet (Lin et al., 2017b). The feature extraction and matching steps are identical to Siamese Faster R-CNN, but it uses

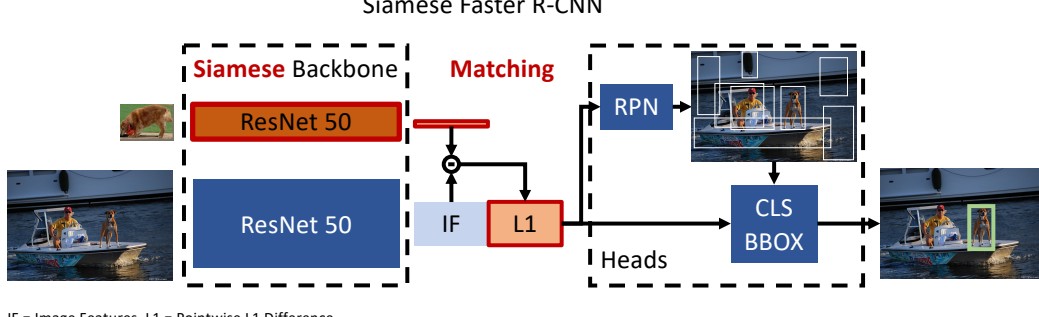

IF = Image Features, L1 = Pointwise L1 Difference,
RPN = Region Proposal Network, CLS = Classifier, BBOX = Bounding Box Regressor

Figure 2: Siamese Faster R-CNN

the unified RetinaHead to jointly propose and classify bounding boxes. To counter the effect of too many negative samples, the classifier is trained with focal loss Lin et al. (2017b).

**Training & Evaluation** During training a reference category is randomly chosen for every image by picking a category with at least one instance in the image. A reference is retrieved by randomly selecting one instance from this category in another image and tightly cropping it. The labels for each bounding box are changed to 0 or 1 depending on whether the object is from the reference category or not. Annotations for objects from the held-out categories are removed from the dataset before training. At test time a similar procedure is chosen but instead of picking one category for each image, all categories with at least one object in the image are chosen Michaelis et al. (2018b) and one (1-shot) or five (5-shot) reference images are provided. Predictions are assigned their corresponding category label and evaluation is performed using standard tools and metrics.

**Implementation** We implemented Siamese Faster R-CNN and Siamese RetinaNet in mmdetection v1.0rc (Chen et al., 2019a), which improved performance by more than 30% over the original paper (Table 4; Michaelis et al., 2018b). We keep all hyperparameters the same as in the standard Faster R-CNN implementation of mmdetection (which achieves 36.4% mAP/58.4% AP$^{50}$ on regular COCO). Due to resource constraints we reduce the number of samples per epoch to 120k for Objects365.

**Datasets** We use the four datasets shown in Table 1: COCO (Lin et al., 2014), Objects365 (Shao et al., 2019), LVIS (Gupta et al., 2019) and Pascal VOC (Everingham et al., 2010). We use standard splits and test on the validation sets except for Pascal VOC where we test on the 2007 test set. Due to resource constraints, we evaluate Objects365 on a fixed subset of 10k images from the validation set. Following common protocol (Michaelis et al., 2018b; Shaban et al., 2017) we split the categories in each dataset into four splits using every fourth category as hold-out set and the other 3/4 categories for training. So on Pascal VOC there are 15 categories for training in each split, on COCO there are 60, on Objects365 274 and on LVIS 902. We train and test four models (one for each split) and report the mean over those four models, so performance is always measured on all categories. Computing performance in this way across all categories is preferable to using a fixed subset as some categories may be harder than others. During evaluation, the reference images are chosen

| Dataset | Version | Classes | Images | Instances | Ins/Img | Cls/Img | Thr. |
|---|---|---|---|---|---|---|---|
| Pascal VOC | 07+12 | 20 | 8,000 | 23,000 | 2.9 | 1.6 | ✓ |
| COCO | 2017 | 80 | 118,000 | 860,000 | 7.3 | 2.9 | ✓ |
| LVIS | v1 | 1,203 | 100,000 | 1,270,000 | $\geq$12.8* | $\geq$3.6* | ✗ |
| Objects365 | v2 | 365 | 1,935,000 | 28,000,000 | 14.6 | 6.1 | ✓ |

Table 1: Dataset comparison. Thr. = Throughtly annotated: every instance of every class is annotated in every image. *LVIS has potentially more objects and categories per image than are annotated due to the non-exhaustive labeling.

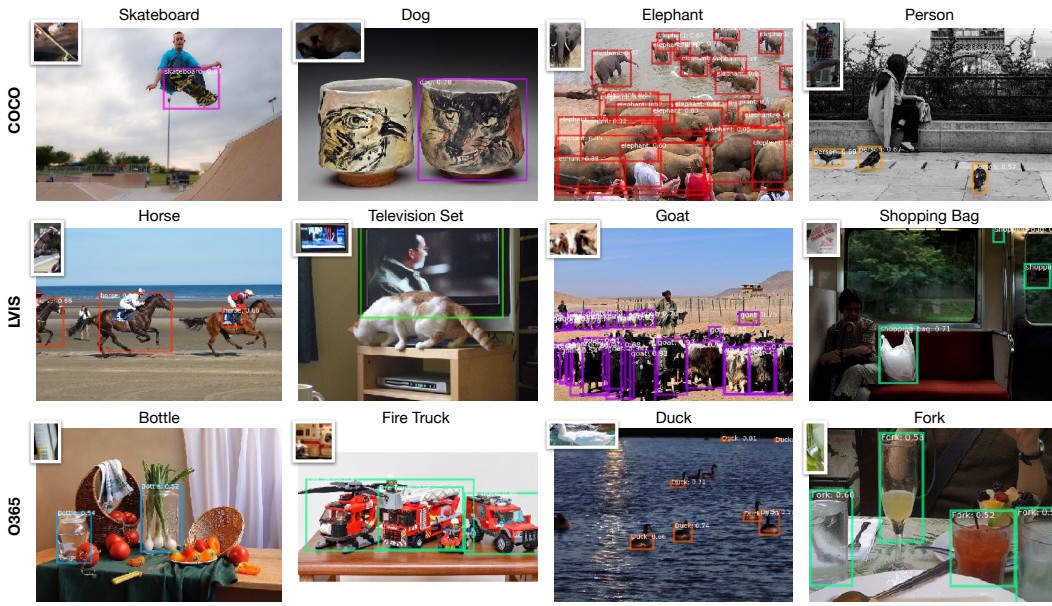

Figure 3: Example predictions on held-out categories (ResNet-50 backbone). The left three columns show success cases. The rightmost column shows failure cases in which objects are overlooked and/or wrongfully detected.

| Model | COCO | | Pascal VOC | |
|---|---|---|---|---|
| | Train Cats. | Held-Out Cats. | Train Cats. | Held-Out Cats. |
| Siamese Faster R-CNN | 49.7 | 22.8 | 82.7 | 37.6 |
| — empty Refs. | 10.1 | 4.4 | 59.6 | 33.2 |

Table 2: On COCO and Pascal VOC there is a clear performance gap (AP$^{50}$) between categories used during training (Train Cats.) and held-out categories (Held-Out Cats.). A baseline getting a black image as reference which contains no information about the target category (– empty Refs.) performs surprisingly well on Pascal VOC but fails on COCO.

randomly. We therefore run the evaluation five times, reporting the average AP$^{50}$ over splits. The 95% confidence intervals for the average AP$^{50}$ is below $\pm 0.2\%$AP$^{50}$ for all experiments.

## 4 RESULTS

### 4.1 GENERALIZATION GAP ON COCO AND PASCAL VOC

We start by showing that objects of held-out categories are detected less reliably on COCO and Pascal VOC. On both datasets, Siamese Faster R-CNN shows strong signs of overfitting to the training categories (Table 2): performance is much higher than for categories held-out during training (COCO: $49.7 \rightarrow 22.8$ %AP$^{50}$; Pascal VOC: $82.7 \rightarrow 37.6$ %AP$^{50}$). We refer to this drop in performance as the *generalization gap*. This result is consistent with the literature: Hsieh et al. (2019) – the previous state-of-the-art – report performance dropping $40.9 \rightarrow 22.0$ %AP$^{50}$ on COCO (see Table 4 below). Some newer models reportedly close the gap on Pascal VOC (Zhang et al., 2019; Hsieh et al., 2019; Li et al., 2020); we will discuss Pascal VOC further in the next section. Example predictions show good localization (bounding boxes) even for unknown objects in cluttered scenes while classification errors make up the majority of mistakes (Fig. 3).

### 4.2 PASCAL VOC IS TOO EASY TO EVALUATE ONE-SHOT OBJECT DETECTION MODELS

Having identified this large generalization gap, we ask whether the models have learned a useful metric for one-shot detection at all or whether they rely on simple dataset statistics. Pascal VOC

contains, on average, only 1.6 categories and 2.9 instances per image. In this case, simply detecting all foreground objects may be a viable strategy. To test how well such a trivial strategy would perform, we provide the model with uninformative references (we use all-black images). Interestingly, this baseline performs very well, achieving 59.6 %AP$^{50}$ on training and 33.2 %AP$^{50}$ on held-out categories (Table 2). For held-out categories, the difference to an example-based search is marginal (33.2 → 37.6 %AP$^{50}$). This result demonstrates that on Pascal VOC the model mostly follows a shortcut and uses basic dataset statistics to solve the task.

In contrast, COCO represents a drastic increase in image complexity compared with Pascal VOC: it contains, on average, 2.9 categories and 7.3 instances per image. As expected, in this case the trivial baseline with uninformative references performs substantially worse than the example-based search (training: 49.7 → 10.1 %AP$^{50}$; held-out: 22.8 → 4.4 %AP$^{50}$; Table 2). Thus, the added image complexity forces the model to indeed rely on the learned metric for identifying matching objects, but this metric does not generalize well.

### 4.3 TRAINING ON MORE CATEGORIES REDUCES THE GENERALIZATION GAP

We now turn to our main hypothesis that increasing the number of categories used during training could close the generalization gap identified above. To this end we use Objects365 and LVIS, two fairly new datasets with 365 and 1203 categories, respectively (much more than the 20/80 in Pascal VOC/COCO). Indeed, training on these wider datasets improves the relative performance on the held-out categories from 46% on COCO to 76% on Objects365 and up to 89% on LVIS (Fig. 4). In absolute numbers this means going from a 26.9 %AP$^{50}$ gap on COCO to a 4.6 %AP$^{50}$ gap on Objects365 and a 3.5 %AP$^{50}$ gap on LVIS (Table 3) in the one-shot setting. Increasing the number of references to five (5-shot) improves performance on all datasets but leaves relative performance unchanged (Table 3, right columns).

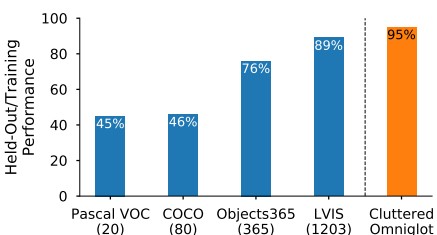

Figure 4: Relative performance grows with the number of categories but stays lower than that of Siamese-U-Net on Cluttered Omniglot (Michaelis et al., 2018a).

This effect is not caused simply by differences between the datasets, as the following experiment shows. For both datasets (LVIS and Objects365), we train models on progressively more categories. When training on less than 100 categories (resembling training on COCO), a clear generalization gap is visible on both LVIS and Objects365 (Fig. 5A: light blue vs. dark blue). Performance on the held-out categories increases with the number of training categories, while performance on the training categories stays the same (LVIS) or decreases (Objects365). The same effect can be seen in the 5-shot setting but with a better baseline performance (Fig. A.2 in Appendix).

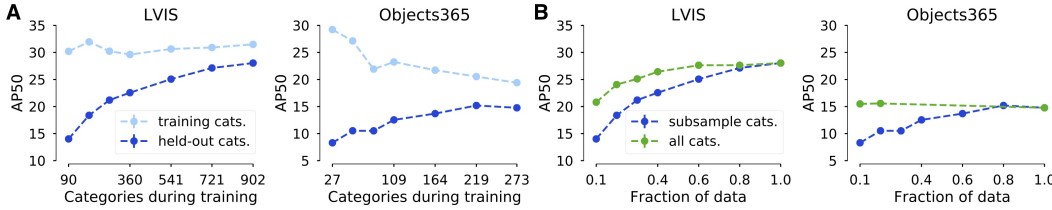

Figure 5: **A.** Experiment subsampling LVIS and Objects365 categories during training. When more categories are used during training performance on held-out categories (blue) improves while performance on the training categories (light blue) stays flat or decreases. **B.** Comparison of the performance on held-out categories if a fixed number of instances is chosen either from all categories (green) or from a subset of categories (blue). Having more categories is more important than having more samples per category. (1-shot results, for 5-shot see Appendix Fig. A.2)

**COCO**

| Model | Backb. | Sched. | 1-shot | | | 5-shot | | |
|---|---|---|---|---|---|---|---|---|
| | | | Train C. | Held-Out C. | Delta | Train C. | Held-Out C. | Delta |
| S-Retina | R50 | 1x | 50.6 | 18.9 | 31.7 | 55.5 | 22.1 | 33.4 |
| S-FRCNN | R50 | 1x | 49.7 | 22.8 | 26.9 | 54.9 | 27.6 | 27.3 |
| S-FRCNN | R50 | 3x | 51.7 | 21.9 | 29.8 | 57.6 | 26.7 | 30.9 |
| S-FRCNN | X101 | 1x | 56.4 | 23.5 | 32.9 | 61.9 | 28.6 | 33.3 |

**LVIS**

| Model | Backb. | Sched. | 1-shot | | | 5-shot | | |
|---|---|---|---|---|---|---|---|---|
| | | | Train C. | Held-Out C. | Delta | Train C. | Held-Out C. | Delta |
| S-Retina | R50 | 1x | 28.4 | 24.7 | 3.7 | 31.6 | 27.5 | 4.1 |
| S-FRCNN | R50 | 1x | 31.5 | 28.0 | 3.5 | 37.0 | 33.0 | 4.0 |
| S-FRCNN | R50 | 3x | 32.7 | 28.7 | 4.0 | 38.2 | 33.5 | 4.7 |
| S-FRCNN | X101 | 1x | 35.4 | 31.3 | 4.1 | 41.4 | 36.3 | 5.1 |

**Objects365**

| Model | Backb. | Sched. | 1-shot | | | 5-shot | | |
|---|---|---|---|---|---|---|---|---|
| | | | Train Cats. | Held-Out C. | Delta | Train C. | Held-Out C. | Delta |
| S-Retina | R50 | 1x | 19.7 | 14.5 | 5.2 | 23.4 | 17.2 | 6.2 |
| S-FRCNN | R50 | 1x | 19.4 | 14.8 | 4.6 | 25.7 | 19.9 | 5.8 |
| S-FRCNN | R50 | 3x | 22.0 | 16.5 | 5.5 | 27.7 | 20.9 | 6.8 |
| S-FRCNN | X101 | 1x | 25.0 | 17.9 | 7.1 | 30.6 | 22.4 | 8.2 |

Table 3: Effect of a three times longer training schedule and a larger backbone (ResNeXt-101 32x4d) on model performance across datasets. While larger models and longer training times lead to no or only minor improvements on held-out categories on COCO, they do have a larger effect on LVIS and Objects365.

### 4.4 THE NUMBER OF CATEGORIES IS THE CRUCIAL FACTOR

The results so far show that increasing the number of categories used during training reduces the generalization gap and improves performance. However, this effect could also be caused by the fact that with more categories there is also more data available. Consider the situation where we train on 10% of the categories (90 in the case of LVIS). As we sample these categories uniformly from the dataset, we use only approximately 10% of the total number of instances. To control for this confound, we created training sets that match the number of instances: in this case we use only 10% of the instances in the dataset but sample them uniformly from all 900 training categories.

The results can be seen in Fig. 5B. Our example from above with 10% of the data corresponds to the leftmost datapoint in both plots. The model trained with more categories (green) clearly outperforms the model with more instances per category (blue). The same performance gap can be seen for any fraction of the data. Thus, for a given budget of instances (labels) it is better to cover more categories than to collect as many samples per category as possible. We will discuss the implications of this result in in Section 5.1.

### 4.5 ON LARGER DATASETS STANDARD TRICKS BENEFIT KNOWN AND NOVEL CATEGORIES (ALMOST) ALIKE

If models indeed learn the distribution over categories, stronger models that can learn more powerful representations should perform better on known and novel categories alike. We test this hypothesis in two ways: first, by replacing the standard ResNet-50 (He et al., 2016) backbone with a more expressive ResNeXt-101 (Xie et al., 2017); second, by using a three times longer training schedule.

The larger backbone does not improve performance on the held-out categories on COCO (Table 3). Instead the additional capacity is used to memorize the training categories, which is evident from the large improvement $(6.7\,\%\text{AP}^{50})$ in performance on the training categories, but only a small improvement $(0.7\,\%\text{AP}^{50})$ on the held-out categories. In contrast, on LVIS and Objects365 the gains of the bigger backbone are not confined to the training categories but applies to the one-shot setting to a much larger extent than on COCO.

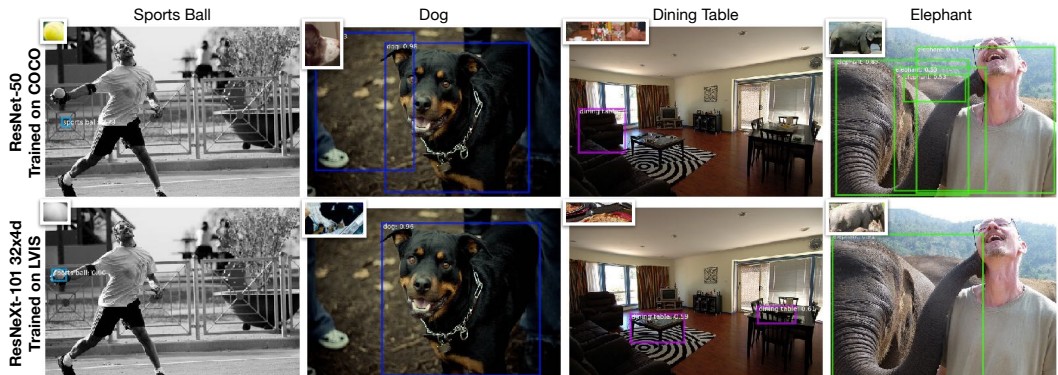

Figure 6: Predictions on COCO tend to be more accurate and cleaner when using a bigger backbone and training on LVIS. Especially on categories with more ambiguous references like sports ball or dining table the LVIS trained model is more precise. Additionally the ResNeXt backbone leads to "cleaner" results with less false positives.

Longer training schedules show the same pattern of results as the larger backbone. For COCO, performance on the training categories improves while performance on held-out categories even gets a bit worse on a 3x schedule (Table 3). In contrast, performance on LVIS and Objects365 improves for both training and held-out categories alike, suggesting that the models do not overfit only the training categories. Note that we do observe a slightly larger improvement on the training categories than on the held-out ones; we will come back to this point in Section 5.1.

## 4.6    RESULTS HOLD FOR DIFFERENT MODEL CONFIGURATIONS

To test if our findings apply to single-stage detectors as well, we train and test Siamese RetinaNet on COCO, LVIS and Objects365 (Table 3). Results are very similar to Siamese Faster R-CNN. Siamese RetinaNet shows a slightly larger generalization gap on COCO (relative performance: Retina: 37% vs. FRCNN: 46%) but results are very similar on LVIS (Retina: 87% vs. FRCNN: 89%) and Objects365 (Retina: 74% vs. FRCNN: 76%).

Taken together our results hold for single- and two-stage detectors with different backbones and learning rate schedules on two datasets (Objects365 and LVIS) for 1-shot and 5-shot evaluation. These results suggests that our conclusions may extend to most object detection models and we can expect to significantly boost performance using the large toolboxes which exist for traditional object detection. A demonstration on COCO is described in the following section.

## 4.7    STATE-OF-THE-ART ONE-SHOT DETECTION ON COCO USING LVIS ANNOTATIONS

Using the insights from above, we now demonstrate state-of-the-art one-shot detection performance on COCO by training on a large number of categories. We use LVIS and create four splits which leave out all categories that have a correspondence in the respective COCO split. As LVIS is a re-annotation of COCO, this means that we expand the categories in the training set while training on the same set of images.

Training with the more diverse LVIS annotations leads to a noticeable performance improvement from 22.8 to 25.0 %AP$^{50}$, which can be improved even further to 27.4 %AP$^{50}$ by using the stronger ResNeXt-101 backbone, outperforming the previous best model by 5.4 %AP$^{50}$ (Table 4). In relative terms that means going from 45% relative performance to 65%, thus substantially outperforming the previous best method (55% relative performance Hsieh et al. (2019)) both in absolute and relative terms. Visual inspection of the results (Appendix Fig. 6) shows that the improved model generates cleaner predictions with less false positives especially for difficult reference images.

| Model | Backb. | Train Data | 1-shot | | 5-shot | |
|---|---|---|---|---|---|---|
| | | | Train C. | Held-Out C. | Train C. | Held-Out C. |
| Siam. Mask R-CNN* | R50 | COCO | 37.6 | 16.3 | 41.3 | 18.5 |
| CoAE† | R50 | COCO | 40.9 | 22.0 | - | - |
| Siam. RetinaNet | R50 | COCO | 50.6 | 18.9 | 55.5 | 22.1 |
| Siam. Faster R-CNN | R50 | COCO | 49.7 | 22.8 | 54.9 | 27.6 |
| Siam. Mask R-CNN | R50 | COCO | 51.9 | 22.9 | 57.9 | 27.8 |
| Siam. Cascade R-CNN | R50 | COCO | 50.3 | 22.0 | 56.2 | 27.2 |
| Siam. Faster R-CNN | X101 32x4d | COCO | **56.4** | 23.5 | **61.9** | 28.6 |
| Siam. Faster R-CNN | R50 | LVIS | 36.2 | 25.0 | 43.5 | 31.7 |
| Siam. Faster R-CNN | X101 32x4d | LVIS | 42.5 | **27.4** | 50.3 | **34.8** |

Table 4: Performance ($AP^{50}$) on COCO can be improved by training on LVIS. Siamese Mask R-CNN and Siamese Cascade R-CNN are identical to Siamese Faster R-CNN except for an additional mask head or cascaded bbox heads. (*Michaelis et al. (2018b), † Hsieh et al. (2019))

## 5 DISCUSSION

### 5.1 FUTURE DATASETS SHOULD FOCUS ON THE DIVERSITY OF CATEGORIES.

Our findings have important implications for the design of future datasets. A broader range of categories is helpful at any dataset size (Fig. 5, right). More importantly, from a certain point onwards more examples per category lead to diminishing returns in terms of generalization. This result is particularly evident in the experiment where we subsample instances versus categories (Fig. 5B): using all the available categories, we need only $\approx 60\%$ of the data on LVIS to achieve optimal performance. On Objects365 this effect is even more extreme: with only 10% of the instances, performance is already saturated. We therefore suggest that future data collection and annotation efforts should focus on a broader set of categories while the number of instances for each of those categories does not have to be as large as e.g. in Objects365.

Despite being a big step forward, models trained on LVIS still show a generalization gap and this gap widens when using stronger models. In this case, increased capacity aids performance on known categories slightly more than on novel categories, suggesting that some amount of overfitting on the training categories occurs. However, the curves for subsampled categories (Fig. 5) are not yet saturated at the maximal number of categories in each dataset, which leaves hope that the gap can be closed by further increasing the number of categories in future datasets.

### 5.2 OUTLOOK

Our insight that applying existing methods on larger and more diverse datasets can lead to unexpected capabilities is mirrored in other areas. This phenomenon has been observed time and again and was termed the "unreasonable effectiveness of data" (Halevy et al., 2009; Sun et al., 2017) or the "bitter lesson" (Sutton, 2019). It played a key role in the breakthrough of DNNs thanks to ImageNet (Russakovsky et al., 2015; Krizhevsky et al., 2012) as well as recent results on gampe-play (Berner et al., 2019) or language modelling (Brown et al., 2020). Recently Kolesnikov et al. (2019) achieve impressive few-shot learning performance including the first approach at one-shot ImageNet. As in our study, using a simple method (transfer learning) on a large and diverse dataset led to results that are far better than what one would have expected: AlexNet performance with 3 samples per class in their case; 89% relative performance on LVIS in our case. We hope that by building on this insight we can soon move from trying to solve few-shot learning towards using few-shot learning to solve other problems.

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

# A APPENDIX

## A.1 HYPERPARAMETER SETTINGS

Our model is derived from mmdetection v1.0rc Chen et al. (2019a) and uses the same hyperparameters as used for Faster R-CNN and RetinaNet. All details can be found in the config files of the respective mmdetection checkpoint: `https://github.com/open-mmlab/mmdetection/tree/5bf935e1b7621b234ddb34ef6c32b2b524243995/configs`. Please note that the default settings for Pascal VOC differ slightly from those for COCO training. We use the COCO hyperparameters for experiments on COCO, LVIS and Objects365 and Pascal VOC settings for Pascal VOC and GroZi-3.2k.

## A.2 GROZI-3.2K

GroZi-3.2k is a dataset of in-store products developed by Osokin et al. (2020). In contrast to the datasets we study in the main paper, the depicted products cover only a very limited range of objects and categories are very fine-grained (one specific product makes one category). Studying such a different dataset will give us a better idea to which scenarios we can expect our findings to generalize.

**Experiment details:** We train Siamese Faster R-CNN with the same hyperparameters as on Pascal VOC but increase the number of epochs to 80 without pre-training and 20 with pre-training. The learning rate step is scheduled after 3/4 of the epochs (60 and 15 epochs respectively). We train on the train split defined in Osokin et al. (2020) and test on `val-old-cl` and `val-new-cl`.

Different to our approach on the other datasets, GroZi-3.2k has a fixed set of reference images which we use for training and evaluation. We therefore cannot average over multiple evaluation runs to determine how much the selection of reference images influences the result. As the validation sets are very small (60 images in `val-old-cl` and 84 in `val-new-cl`) we can assume that the error is rather large compared to the 0.2 % $AP^{50}$ we observe on the other datasets.

Note that our results are not directly comparable to those presented in Osokin et al. (2020) as our evaluation procedure is markedly different. Most importantly we don't ignore 'difficult' examples which explains much of the drop in performance we observe.

**Results:** Without pre-training we find a significant generalization gap (Table 5, first row). As for LVIS and Objects365 performance on held-out categories depends more on the number of categories than the amount of data alone (Fig. A.1A).

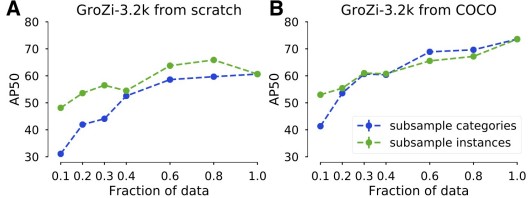

Figure A.1: Performance on GroZi3.2 (Osokin et al., 2020) held-out categories.

However, small datasets are usually not used to train models from scratch but most approaches will involve some form of transfer learning. We therefore train and tested models that were pre-trained on COCO and LVIS. These models don't show a generalization gap (Table 5, Pre-Train COCO & LVIS) and performance depends more or less only on the amount of data available (Fig. A.1B).

**Conclusion:** In contrast to the other datasets we study GroZi-3.2k has a very fine-grained set of categories from a narrow range of objects (in store products). So it is pretty remarkable that we find very similar effects when training from scratch. In contrast the results with pre-training may

| Model | Pre-Train | val-old-cl | val-new-cl |
|---------|-----------|------------|------------|
| S-FRCNN | - | 81.7 | 60.6 |
| S-FRCNN | COCO | 74.9 | 73.6 |
| S-FRCNN | LVIS | 75.8 | 69.1 |

Table 5: Performance ($AP^{50}$) on GroZi-3.2k

at first seem to contradict our previous findings. They however make sense when compared to the situation on COCO: There training on the more diverse LVIS dataset leads to better results and a smaller generalization gap. We here have a very similar scenario where we pre-train on much larger and more diverse datasets and consequently find better generalization capabilities. This is also very similar to the findings of Kolesnikov et al. (2019) who achieve good results for few-shot learning on ImageNet by training on a significantly larger dataset (Section 5.2).

## A.3 ADDITIONAL FEW-SHOT RESULTS

We provide five-shot results for the experiments in Fig. 5 in Fig. A.2.

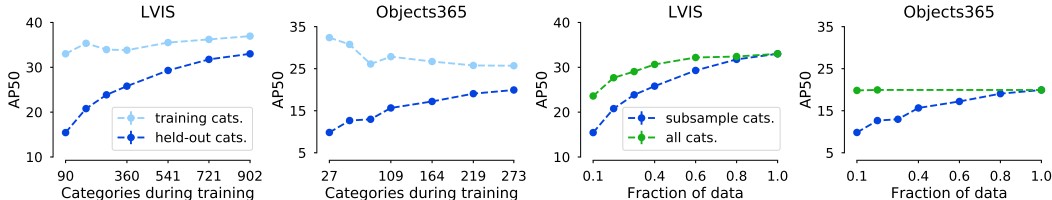

Figure A.2: Five-shot results for the subsampling experiments in Fig. 5.

