# OpenReview forum: "Closing the Generalization Gap in One-Shot Object Detection"
_ICLR.cc/2021/Conference — Reject_

### Official Review · AnonReviewer1 · 2020-10-24
**In this paper, authors propose to close the generalization gap in one-shot object detection by increasing the number of object categories used in training. The paper also showed that standard methods to improve object detection models like stronger backbones or longer training schedules also benefit novel categories, which was not the case for smaller datasets. The conclusion provides guidelines for future data collection.**

**Rating:** 7
**Confidence:** 4

**Review:**

Strengths:
- The paper is well written and it’s easy to follow the story.
- It provides a comprehensive review on related papers on object detection especially one-shot detection and their limitations.
- The idea is simple and clear. Although the paper focuses on the data used for training, it gave great insights for understanding the generalization of object detector and provides practical guidelines for future large scale data collection.
- Finally the extensive experimentation and analysis on results are convincing.

Weaknesses:
- The paper studies the importance of number of object categories in training dataset and claims that the gap in one-shot detection can be closed by increasing the number of categories. However, it’s not the case that the more categories the better, there should also be enough diversity in data distribution and granularity in label definition. This is also an important guideline for future data collection. It would be interesting to see some analysis on data diversity and label granularity.

Minor comments:
- Fig 4 was not referred anywhere in the paragraphs.
- In Table 3, the results will be complete with one more experiment using X101 and 3x are together
- Typo in Figure 5’s caption: “ether” -> either

---

> ### Author Response · Authors · 2020-11-12
> **We'll add a discussion of data diversity and granularity!**
>
> Thank you for your kind review! We are happy you liked the paper. We agree that the diversity and granularity of the data play an important role and add a corresponding discussion to the paper. It would be great to provide clear data on this but we are not sure if there is any specific experiment to test the influence of diversity and granularity that can be realized without collecting a new dataset. If you have any ideas please let us know!

---

> > ### Comment · Area_Chair1 · 2020-11-12
> > **suggesting a dataset with a different granularity-diversity regime**
> >
> > To somehow address the question of granularity-diversity of categories, I suggest trying the method on the data from this paper: "A. Osokin, D. Sumin, V. Lomakin. OS2D: One-Stage One-Shot Object Detection by Matching Anchor Features, In ECCV 2020"
> > It uses a different from the standard benchmarks regime (more classes and fewer instances per class, narrower definition of a class).

---

> > > ### Author Response · Authors · 2020-11-13
> > > **good suggestion**
> > >
> > > We will try to reproduce Figure 5 (number of category/performance trade off) with this dataset!

---

### Official Review · AnonReviewer4 · 2020-10-27
**I appreciate the proposed hypothesis, but my major concerns are the experiments. Please refer to my comments.**

**Rating:** 6
**Confidence:** 3

**Review:**

Pros.:
1. The hypothesis is interesting and novel
2. The paper is clearly presented and well-organized.
3. The authors conduct many ablation studies to validate the proposed hypothesis.

Cons:
1. Claimed observation:

In your claimed observation, increasing the number of classes could improve performance. However, increasing the number of classes also increasing the number of training samples in your experiments, and using more training data could have higher performance. Therefore, if the total number of training images is fixed, what are the results when more classes are included?

2. About the usage of Siamese Faster R-CNN

The siamese faster R-CNN is a suitable network to validate the hypothesis, but it is somewhat old even with a more powerful backbone network. If more powerful methods are used, what is the performance gap? If more powerful methods could reduce the performance gap more, I think the performance issue is not related to the number of classes. The authors should adopt the latest method, such as [Ref.1,2,3], to repeat the experiments.

[Ref. 1] Hsieh et al., One-Shot Object Detection with Co-Attention and Co-Excitation, NIPS'19

[Ref. 2] Osokin et al., OS2D: One-Stage One-Shot Object Detection by Matching Anchor Features, ECCV'20

[Ref. 3] Li et al., One-Shot Object Detection without Fine-Tuning, arXiv' 20

3. About the shot number

The authors focus on only the one-shot setting, but the proposed hypothesis could be kept for the N-shot setting where N > 2.
If more shots are used, what is the performance gap? Besides, with more shots, could the proposed hypothesis be still effective? The authors should conduct the same experiments under a different number of shots.

Overall, I appreciate the proposed hypothesis, but my primary concerns are the experiments. Please refer to my comments above.

---

> ### Author Response · Authors · 2020-11-12
> **Clarification and adding suggested methods**
>
> Thank you for your review! We want to quickly address your two central concerns.
>
> The first concern is that increasing the number of classes also increases the number of training samples which could explain the effect. We tested exactly this scenario in Figure 5B comparing subsets with the same number of instances but from a fixed number of classes (blue) and all classes (green). We should have made this clearer in the text and will update the manuscript accordingly.
>
> The second concern is that newer methods than Siamese Faster R-CNN may already have solved the generalization issue. We think this is a very important point and addressing it will make the paper stronger. Of the suggested models only CoAE from Hiseh et. al. evaluate on COCO improving the relative performance of Siamese Faster R-CNN from 45% to 55%. However Siamese Faster R-CNN trained on LVIS achieves 65% relative performance on COCO and 89% on LVIS. We nevertheless think it is a good idea to repeat our experiments with more models but cannot promise to adapt, train and test all three suggested methods within the two week discussion period. A somewhat realistic option would be to add a Siamese version of RetinaNet to also cover single stage detectors as well as Hsieh et al. 2019 which uses the same task setup as we do. Would adding the above discussion to the manuscript as well as the results for these two models be sufficient to address your concern?

---

> > ### Comment · AnonReviewer4 · 2020-11-14
> > **Thanks for the response**
> >
> > Thanks for the response.
> >
> > I will be glad to see the responses to the first and second concerns. These two responses seem good.
> >
> > For the first concern, I am sorry that I don't see the results clearly, but you could make the paper more clear.
> >
> > For the second concern, the discussion the authors mention is enough, and please add it to the manuscript.
> >
> > Besides, where is the response to the third concern? Please address it if possible.

---

> > > ### Author Response · Authors · 2020-11-14
> > > **Addressing the number of shots**
> > >
> > > Thanks for you reply! We will make the paper clearer with regard to your first concern and add the discussion about the second. We had planned to address all comments from you and all the other reviewers once the most important points are clarified but failed to mention this in our initial answer. Sorry.
> > >
> > >
> > > Now to address your question about the number of shots:
> > > We are happy to rerun the analysis with more shots. While we expect results with N-shots to be very similar to the one-shot setting we agree that adding them would strengthen our hypothesis. Regarding the choice of N we would suggest using N=5 to match the setting in Michaelis et. al. 2018 but are happy to use any other number if you think it would be more informative.

---

> > > > ### Comment · AnonReviewer4 · 2020-11-15
> > > > **Thanks for the response to the 3rd comments**
> > > >
> > > > That's okay.
> > > >
> > > > Thanks for your response to this comment, and now I don't have other comments now.

---

### Official Review · AnonReviewer3 · 2020-10-28
**Further study is needed to verify the generalization ability of the main claims.**

**Rating:** 6
**Confidence:** 5

**Review:**

This paper provides a variety of studies to understand the generalization gap between known and novel classes in one-shot object detection. The studies are carried out by using siamese Faster R-CNN framework on four benchmark datasets. The most notable observation was that it was more important to increase the number of object category than to increase the number of instances per each category in order to reduce the generalization gap. This observation is very useful to anyone planning to build a dataset for this task or implement the appropriate method. Figure 5 is very important and well presented to support the main claim.

However, there are some factors that need further studies to fully trust this observation in terms of generalization capabilities:
1. Is it possible to get the same observations from other one-shot object detection methods other than the siamease Faster R-CNN?
2. Can this observation be presented from using a variety of backbone methods which can be either a shallow method or a deeper CNN model?
3. Can this claim be applied to any kind of category? (e.g., detection of person category having a more diverse appearances can be more affected by the number of instances used in training.)

In addition, why did increasing the number of categories reduce the detection accuracy of known objects? (Figure 5A) Did the method suffer from training as the number of categories increased? In other words, do we need to improve the accuracy of novel object classes at the expense of reduced detection accuracy of known classes?

---

> ### Author Response · Authors · 2020-11-12
> **Adding additional models**
>
> Thank you for your review! We want to quickly address your central concern and would be grateful for a quick feedback. We are confident that our findings will also hold for other models and backbones, and we specifically chose to build upon Faster R-CNN because it is widely used in object detection. Nevertheless having data on this question would definitely be interesting and make the paper stronger. We are happy to train and test a few additional models. Do you think adding a model based on a single stage detector as well as the newer model from Hsieh et al. 2019 would be sufficient to make the results trustworthy?

---

> > ### Comment · Area_Chair1 · 2020-11-12
> > **baseline suggestion**
> >
> > As far as I understand, the work of Hsieh et al. 2019 is also a two stage method (but with region proposals depending on the class-to-detect).
> > To the best of my knowledge, the current SOTA is "Q. Fan, W. Zhuo, C.-K. Tang, and Y.-W. Tai. Few-shot object detection with attention-rpn and multi-relation detector. In CVPR, 2020." and it would be great to see if the results hold for their method.

---

> > > ### Author Response · Authors · 2020-11-13
> > > **thanks for the suggestion**
> > >
> > > Fan. et. al. is indeed newer and more versatile so we will look at it first and return to Hsieh et. al. later if enough time is left.

---

> > > > ### Comment · AnonReviewer3 · 2020-11-25
> > > > **Most of my concerns remain.**
> > > >
> > > > Thank you for addressing the first and second questions. I recommend adding experiments performed with variety of models to support that key observation can be applied to any backbone model and object detection framework.
> > > >
> > > > I am still waiting for authors' response for the third question. Some categories have non-rigid appearance so it may require many images to capture different appearance instances in training these categories.

---

> > > > > ### Author Response · Authors · 2020-11-25
> > > > > **Why do your concerns still remain?**
> > > > >
> > > > > We added another model, dataset and 5-shot evaluation to a total of 5 datasets, 2 models, 2 backbones, 2 learning rate schedules and two evaluation settings. Our findings are consistent across all these settings and agree with concurrent findings in [1] and [2] (See Sections 2 & 5.2). We are not really sure which additional models we missed that can be expected to substantially change the picture?
> > > > >
> > > > > Regarding the third question: Looking at the effect on different categories is an interesting question. However, it is highly non-trivial: in addition to rigidity – which you mention – object size, typical location, indoor/outdoor objects, granularity (stool vs. bar stool) etc. all play into this as well. Trying to separate these issues is definitely interesting, but we consider it out-of-scope for the current paper due to its high complexity
> > > > >
> > > > > [1] Jiang et. al. 2020, “Dataset Bias in Few-shot Image Recognition” \
> > > > > [2] Fan et. al. 2020, “Few-Shot Object Detection with Attention-RPN and Multi-Relation Detector”

---

### Official Review · AnonReviewer2 · 2020-10-30
**Official Blind Review #3**

**Rating:** 5
**Confidence:** 4

**Review:**

The paper suggests that a major factor for increasing few-shot performance in the few-shot object detection task is the number of categories in the base training set used to pre-train the few-shot model on a large set of data before it is adapted to novel categories using only a few (or even 1) examples. This effect is measured by the authors by trying out the existing Siamese few-shot detector on 4 datasets: PASCAL, COCO, Objects365, and LVIS showing that the gap in performance on the seen training and the unseen (novel) testing categories is reduced when the base dataset has more classes (e.g. on LVIS where there are more than 1K classes, this "generalization" gap is shown to be minimal). The authors also quantify empirically the effect of increasing the model size and of prolonging the training schedule on this gap. As well as testing on COCO classes while training on LVIS.

Pros:
- number of base classes is indeed an important factor in few-shot methods performance (not just in detection)
- the paper is easy to follow and generally well written, the message conveyed is clear and the experiments are useful

Cons:
- the positive effect of increasing the number of base classes on few-shot performance is long since known, and numerous works even in the few-shot classification literature have noted this fact, so nothing seems to be new here
- the effect of increasing backbone size and prolonging the train schedule does not seem to indicate a strong correlation to the number of train classes, the original gap is maintained, while the gains of the tested modifications seem to be relatively similar up to some noise
- I might be wrong, but it seems the authors mostly target few-shot localization, assuming the target object (given by the reference image example) is always present in the image. This is opposed to what I understand by few-shot detection, wherein a test episode there are several target objects, each accompanied by its support example and query images can have an arbitrary mix of these target objects or none at all.

To summarize: I like the paper, yet I fear it does not meet the plank of what I would consider a paper fitting ICLR standards in terms of novelty. There is nothing wrong in not proposing a new algorithm and instead - uncovering an important fact that was so far overlooked, but as I noted above this is not the case, the paper highlights a well known fact.... I would be happy to monitor other reviewers responses and authors comments on this issue of novelty and would be happy to be convinced otherwise.

---

> ### Author Response · Authors · 2020-11-12
> **Investigating the category/instance tradeoff is a key novelty**
>
> Thank you for your review! We want to quickly address your central concern, the novelty of our findings: We agree that it is not entirely unexpected that larger datasets and more categories improve few-shot generalization. However, we are not aware of any studies that systematically investigated which dimension of the dataset size is more important: the number of categories or the number of instances? We find that the number of categories is way more important than the number of instances. For example, performance on Objects365 is unaffected when using only 10% of the instances of each class (Fig. 5B). We believe that if this result was common sense, the creators of Objects365 would probably have prioritized annotating more classes over annotating that many instances. Thus, could you clarify which studies specifically you think undermine the novelty of our work?

---

> > ### Comment · AnonReviewer2 · 2020-11-13
> > **number of categories positive effect on few-shot performance common knowledge**
> >
> > I will try to find you explicit references which are not my own,
> > I did see it stated in some papers and even wrote it myself.
> >
> > The thing is it might have been just stated without explicit experiments to back it up.
> > So I would leave it to the others and the AC to decide what to do with this concern.
> > For simple indirect evidence for this please see below:
> >
> > The most trivial long-standing evidence of this is the few-shot classification performance on tiered-ImageNet. The tiered-ImageNet is planned as a much harder dataset than for e.g. mini-ImageNet, as in tiered the train / val / test split is done using coarse categories (according to ImageNet WordNet hierarchy), so the fine classes comprising these splits are very different (in mini these are just random).
> > And yet, in all performance tables, you would see tiered has significantly higher numbers then mini, why is that?
> > Well because of the number of categories ofc!
> > Mini has only 60 categories in its train set, while tiered has 351!
> > Having more train categories induces a more diverse set of attributes retained by the model (it has to do it as it needs to support a diverse set of categories) and this leads to more attributes that 'fire' on the test classes (different as they are) and can be used to separate them.
> >
> > Now this intuition I have for quite a long time, and even put it in some papers (in general discussion style),
> > so I was a bit surprised you decided to make a paper out of it :-)
> > But again, if other reviewers and the AC feel there should be one more paper stating this more explicitly and with specific experiments, then by all means - enjoy! :-)

---

> > > ### Author Response · Authors · 2020-11-14
> > > **one of our key contributions is to turn the positive effect of more categories from an intuition into a finding**
> > >
> > > TieredImageNet vs miniImageNet is an excellent example that underscores our point and we will discuss it in the paper. There are three main differences between the two datasets: The number of classes (60 vs. 608), the number of images (60k vs 779k) and the different class structure. So while we share your intuition, we would like to stress that it is only a hypothesis. To establish that the number of classes is indeed the cause of better performance on tieredImageNet, one would have to do precisely the experiment we perform in Fig. 5 of our paper: test performance on a subset of tieredImageNet that contains only 60k images but from 600 classes. However, we are unaware of any paper having done this experiment.
> > >
> > > Thus, one of our key contributions is taking exactly these points apart finding that more classes help predominantly on novel categories (Figure 5, A), that given a fixed budget of instances it is better to have more categories than more samples per category (Figure 5, B) and that these effects are robust across different class structures (curves for LVIS and Objects365 are overall very similar).
> > >
> > > We think that this detailed analysis is an important addition that will help to better understand the key challenges in few-shot learning aiding the design of future datasets, tasks and methods. We will update our contribution and discussion sections to make clearer why this is new and why we consider it to be important.

---

> > > > ### Comment · AnonReviewer2 · 2020-11-22
> > > > **discussion summary**
> > > >
> > > > I agree that additional analysis confirming a (known) intuition is useful.
> > > > (here is one reference making a similar observation among others: https://arxiv.org/abs/2008.07960)
> > > > My only concern is - does it fit the expected ICLR paper grade?
> > > > Here I thought the answer is negative,
> > > > and yet I am not sure and leave it to the other members of the reviewing committee to decide.

---

> > > > > ### Author Response · Authors · 2020-11-25
> > > > > **regarding significance**
> > > > >
> > > > > Thank you for your response. We're happy to see that you agree the paper is solid. Regarding significance: We demonstrate a substantial >5% improvement of the SotA (22 -> 27.5) in few-shot detection and provide solid experiments explaining this improvement. What else does it need to fit the ICLR paper grade?

---

### Author Response · Authors · 2020-11-24
**Model is SotA by a large margin**

We thank the reviewers and AC for the discussion. Your remarks helped to make the paper better and we updated it accordingly. Below we address the two main concerns: 1. is demonstrating the effect of more categories publication worthy (R2) and 2. are the experiments sufficient to verify the effect (R3,R4).


1. Contribution:

We would like to stress that we increase SotA on few-shot detection on COCO by 5.4% points (from 22 to 27.5) and improve generalization from seen to unseen classes from 45% to 89%. These gains are substantial and represent the largest improvements by a single paper in this field we are aware of. We updated the manuscript to make this point clearer.

R2 argues that the intuition that more classes will improve performance is widely shared within the few-shot learning community, and therefore our paper does not present an important enough contribution. If this was indeed the case, why do most papers in few-shot detection still work with Pascal and COCO? Why does a dataset like Objects365 contain so many instances per class with comparably few classes instead of allocating the annotation efforts more efficiently? We agree that intuition is important in guiding the scientific process, but science also requires thorough testing of hypotheses, which is precisely what our paper does – and the results speak for themselves.


\
2. Are the experiments sufficient?:

R3 and R4 both note that we tested only one (comparably simple) model. They ask whether different models may already close the gap (R4) and if not if they at least showed the same behaviour.

These are good questions, and we have performed additional analyses that confirm our  original claims. Here are the short answers:
- Our Siamese Faster R-CNN actually outperforms [1] (suggested by R4, 22.8 vs. 22.0% mAP) on COCO already without pre-training on LVIS and therefore is SotA on this task
- Other models do not close the gap either (e.g. [1]: 22.0/40.9% mAP, i.e.55% relative performance on COCO)
- As suggested by the reviewers and AC, we tested additional an additional model (Siamese RetinaNet), an additional Dataset (GroZi-3.2k) and 5-shot evaluation, showing that our results hold across the following conditions:
  - 5 Datasets - COCO, LVIS, Objects365, Pascal VOC, GroZi-3.2k
  - 2 Models - Siamese Faster R-CNN, Siamese RetinaNet
  - 2 Backbones - ResNet-50 and ResNeXt-101_32x4d
   - 2 Training schedules - 12 epochs (1x) adn 36 epochs (3x)
   - One-Shot & Few-Shot evaluation
   - Transfer evaluation COCO -> LVIS
- Across all these dimensions, training on more categories consistently improves generalization, reduces overfitting and is more important than simply having more data.


\
List of changes:

- For 1. Contribution: We updated the Abstract, the conclusions in the Introduction (Section 1, bullet points), Related Work (Section 2) and the Discussion (Sections 5 & 6)
- For 2. Are the experiments sufficient?: We added the described experiments to Sections 3, 4 and the Appendix. See especially Tables 3 & 4, Section 4.6, Appendix A.2 as well as the point-by-point comments below.
- [R2] Add tiered vs. miniImageNet and [2]: We added a section “Number of categories in few-shot learning” to related work (Section 2)
- [R3, AC] Evaluate additional models: We added results for a single-stage detector, Siamese RetinaNet (Sections 3 & 4.6, Tables 3 & 4). The results confirm our findings with Siamese Faster R-CNN. Despite our best efforts we were not able to add the model suggested by the AC as we ran into severe problems with their implementation.
- [R3] Does the finding hold for different backbones?: Yes, we tested ResNet-50 and ResNeXt-101_32x4d (Section 4.5)
- [R4] Do other methods also close the gap?: No, see discussion above and updated Section 4.1
- [R4] Make clearer that we control for dataset size: We added this to the conclusions in the Introduction (Section 1, bullet point 3) and updated Section 4.4.
- [R4] Evaluate 5-shot: We added 5-shot results to all experiments (Tables 3 & 4, Figure 8)
- [R1, AC] Test on a more fine-grained dataset: We trained and tested Siamese Faster R-CNN on GroZi-3.2k [3] (suggested by the AC). Full details in Appendix A.2.


\
References: \
[1] Hsieh et. al. NeurIPS 2019, “One-Shot Object Detection with Co-Attention and Co-Excitation” \
[2] Jiang et. al. 2020, “Dataset Bias in Few-shot Image Recognition” \
[3] Oskin et. al. ECCV 2020, “OS2D: One-Stage One-Shot Object Detection by Matching Anchor Features”

---

### Decision · Program_Chairs · 2021-01-07
**Final Decision**

**Decision:**

Reject

**Comment:**

The reviewers have ranked this paper as borderline accept. On the negative side, the main claim of the paper (the more categories for training a one-shot detector, the better) has already been observed in several works and very intuitive. However, the paper has done significant experimental work to support this claim. The paper is very well written, it carefully explores the existing setups for one-shot detection and highlights their weaknesses. The paper also gives advice on how to construct better datasets for one-shot detection (the conclusion "add more diverse categories" is somewhat obvious but the paper demonstrates how important that is).